# Waterborne Polyurea Coatings Filled with Sulfonated Graphene Improved Anti-Corrosion Performance

**Jijia Zhang [1], Jihu Wang [1,\*], Shaoguo Wen [1], Siwei Li [1], Yabo Chen [1], Jing Wang [1], Yan Wang [1], Changrui Wang [1], Xiangyi Yu [2] and Yan Mao [2]**

[1] College of Chemistry and Chemical Engineering, Shanghai University of Engineering Science, Shanghai 201620, China; M040118141@sues.edu.cn (J.Z.); sgwen1@sues.edu.cn (S.W.); M040118170@sues.edu.cn (S.L.); M040218132@sues.edu.cn (Y.C.); M040119224@sues.edu.cn (J.W.); M040117117@sues.edu.cn (Y.W.); M040118115@sues.edu.cn (C.W.)

[2] Solid Waste and Chemicals Management Center of Ministry of Ecology and Environment, Beijing 100029, China; yuxiangyi@meescc.cn (X.Y.); maoyan@meescc.cn (Y.M.)

\* Correspondence: wangjihu@sues.edu.cn

**Abstract:** In this paper, an environmentally friendly waterborne polyurea (WPUA) emulsion and its corresponding coating were prepared, which was characterized by dynamic light scattering (DLS), Fourier transform infrared spectroscopy (FTIR), nuclear magnetic resonance (NMR), and scanning electron microscopy (SEM). To improve the performance of the coating, we doped sulfonated graphene (SG) into WPUA to prepare composite coating (SG/WPUA). SG can be uniformly dispersed in WPUA emulsion and is stable for a long time (28 days) without delamination. The water resistance of the composite coating with 0.3 wt.% SG nanofiller was improved; the water contact angle (WCA) result was SG/WPUA (89°) > WPUA (48.5°), and water absorption result was SG/WPUA (2.90%) < WPUA (9.98%). After water immersion treatment, SEM observation revealed that the SG/WPUA film only generated enlarged microcracks (100 nm) instead of holes (150–400 nm, WPUA film). Polarization curves and electrochemical impedance spectroscopy (EIS) tests show that SG nanosheets with low doping content (0.3 wt.%) are more conducive to the corrosion resistance of WPUA coatings, and the model was established to explain the mechanism.

**Keywords:** waterborne polyurea; synthesis; sulfonated graphene; anti-corrosion

## 1. Introduction

Polyurea coatings, especially mainstream commercial 2K-polyureas, have been challengingly applied to high-speed railways, bridges, vehicles, oil and gas pipeline engineering, bulletproof and explosive loads, etc. [1–4]. Due to the extremely rapid kinetics of the isocyanate–amine reaction [5], practical processing of polyurea mandates the use of a reactive spray coating technique. The cost of spray requirements is relatively expensive, and operation is strict, which is not suitable for a wide range of commercial and civilian use. For better workability of polyurea coatings, using more or fewer solvents to adjust the viscosity and volume ratio of raw materials is also unavoidable with regard to environmental pollution. With the vigorous advocacy of environmental protection laws and regulations, alternative water-borne polyurea (WPUA) coatings that are good for construction and solvent-free have attracted widespread attention [6]. However, there are relatively few studies on WPUA coatings.

WPUA is a typical multi-block copolymer; a microphase separation structure exists between the alternately connected soft and hard segments [7]. Its molecular structure contains urea groups, which function as physical crosslinking agents, forming intramolecular and intermolecular hydrogen bonds in the polymer, thereby constructing a physical crosslinking network to induce self-healing properties [8]. In addition, the urea groups give polyurea better quality, such as higher polarity, crystallinity, hardness and melt temperature, etc. [9–11]. Generally, the self-emulsified polyurea is electrostatically stabilized in

water by introducing hydrophilic structural units into the macromolecular chain, but the corresponding coating may have poor water resistance due to its high dispersion in water. Thus, shrinkages, breaks, and internal defects in the coatings caused by water volatilization are inevitable during the coating forming process [12]. These inherent problems will seriously weaken the coatings' resistance to the corrosive medium, resulting in a greatly shortening coating life. Therefore, there is an urgent need to elevate the anticorrosive performance of WPUA coating.

Graphene, as a 2D nanomaterial with high diameter/thickness ratio, perfect water and oxygen resistance, ion impermeability [13], and chemical inertness, has been gradually recognized and exploited [14]. However, the poor compatibility and dispersion of inorganic material incorporated in most polymers is a problem that needs to be solved urgently [15,16], especially the high specific surface area and strong van der Waals force of graphene and its derivatives (graphene oxide, GO) lead to aggregation [16,17]. The covalent functionalization of graphene/GO is usually a more effective method; it has been reported to utilize the coupling agent [18,19], polyol [20], and organic amines [21,22] to modify GO by covalent modification. However, such an approach further destroyed the π-π conjugated network and caused serious structural defects of graphene [23]. To improve the application performance of modified GO, some reducing agents are usually applied to reduce the modified GO [24]. Introducing graphene and its derivatives with "nano-barrier effect" into the organic coating is conducive to forming a "labyrinth effect" inside to prolong the penetration path of the corrosive medium, producing lightweight, low-addition, high-performance anti-corrosion coating [25]. However, there are few studies on doping graphene and its derivatives with WPUA matrix.

Herein, this paper proposes introducing sulfonated graphene (SG) nanosheets into WPUA to improve material properties. In this proposal, SG was prepared by covalently grafting sulfonic acid groups to GO, and then reduction procedure. The SG has excellent water dispersibility with stronger electrostatic interaction and higher dissociation degree of the sulfonic acid group. Additionally, the reduction reaction of the oxygen-containing groups eliminates the surface defects of graphene to a certain extent. In this strategy, SG can be uniformly and stably dispersed in the WPUA emulsion, resulting in a barrier network that can effectively improve the corrosion resistance of the coating. To verify the feasibility of the hypothesis, the interface compatibility and dispersion stability of SG in WPUA emulsion were studied. The water-resistance and corrosion-resistance of composite coatings were systematically evaluated, the optimal mixing ratio of SG was explored, and the corrosion resistance mechanism of the composite coating was studied by establishing a model.

## 2. Experimental

### 2.1. Raw Materials

Graphite powder was obtained from Hefei Microcrystalline Material Technology Co., Ltd. (Hefei, Anhui, China). Sulfanilic acid, hydrochloric acid (37%), hydrazine hydrate (80%), sodium nitrite and acetone (99.5%) were purchased from Adamas Reagent Co., Ltd. (Shanghai, China), 5-amino-1,3,3-trimethylcyclohexanemethylamine (IPDI, cis and trans mixture, 99%), Ditin butyl dilaurate (DBTDL, 95%), 2,2-Bis (hydroxymethyl) propionic acid (DMPA, 98%), Poly (propylene glycol) bis (2-aminopropyl ether) (D2000, average Mn ~2000), Poly (propylene glycol) bis (2-aminopropyl ether) (D230, average Mn ~230), triethylamine (TEA, 99%). Other reagents were purchased from Aladdin Shanghai Co., Ltd. (Shanghai, China). All chemicals are used without further purification.

### 2.2. Preparation of SG

Figure 1 illustrates the schematic of the synthesis of SG. Graphene oxide (GO) was synthesized according to the well-developed Hummers' method [26]. SG was prepared by a diazotization reaction. In total, 55 mg of sodium nitrite was dissolved in deionized water, then 138 mg of p-aminobenzenesulfonic acid and 1 ml of hydrochloric acid aqueous

solution (1 mol/L) were added and stirred to obtain aryl diazonium salt solution at 0–5 °C. The aryl diazonium salt solution was dropped into GO solution in an ice bath under stirring for 2 h and then dialyzed in deionized water to prepare sulfonated graphene oxide (SGO) dispersion. The SGO was reacted with hydrazine hydrate at 98 °C for 12 h to remove the remaining oxygen-containing groups. After centrifugation and repeated washing with deionized water, sulfonated graphene (SG) was obtained.

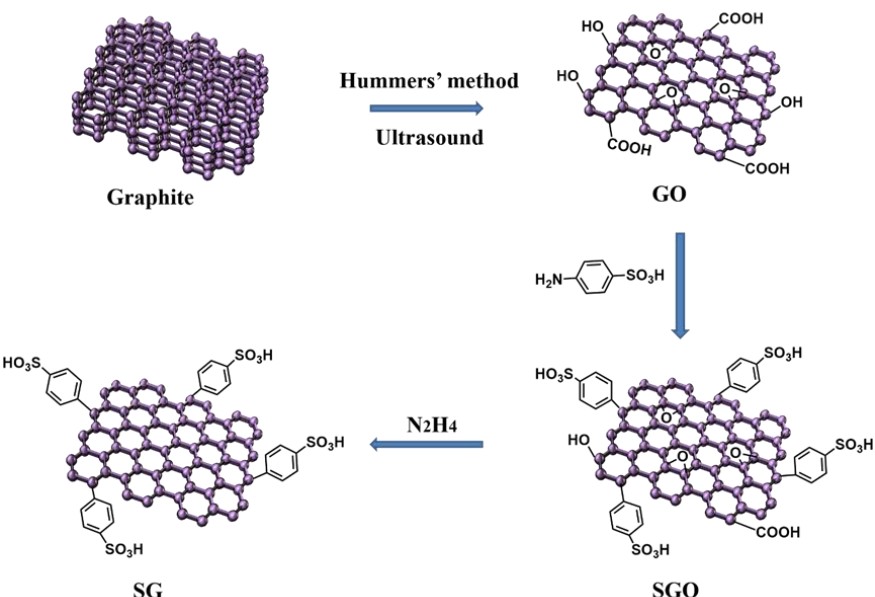

**Figure 1.** Schematic drawing of the synthesis of sulfonated graphene (SG).

### 2.3. Preparation of Waterborne Polyurea

The preparation procedure of waterborne polyurea is shown in Figure 2. First, 17.31 g of IPDI, 2.49 g of DMPA, 2 drops of DBTDL, and a small amount of acetone were added to a 500 mL three-necked flask and stirred in a 50 °C water bath for 4 h under nitrogen atmosphere. Then, 37.09 g of D2000 was added and the reaction was stirred at room temperature for 40 min. Secondly, a mixture of 4.26 g of D230 and 8 mL of acetone was slowly injected into the device using a constant pressure dropping funnel, and the reaction was stirred at room temperature for 40 min. Next, 1.88 g of TEA was added and stirred to neutralize the reaction for 30 min. The product was placed in deionized water, and the WPUA emulsion was prepared by stirring (1500 r) for 30 min. Finally, acetone was removed by distillation under low pressure.

### 2.4. Preparation of Composite Coatings

The sulfonated graphene/water-borne polyurea (SG/WPUA) composite coating was prepared by uniformly dispersing SG in the polyurea emulsion. Based on the weight of pure resin, the addition amount of SG is 0.1 wt.%, 0.3 wt.%, 0.5 wt.%, 0.7 wt.%, and named as 0.1-SG, 0.3-SG, 0.5-SG, 0.7-SG, respectively. According to the above method, graphene oxide/water-borne polyurea (GO/WPUA) composite coatings were also prepared and named. The water-borne polyurea coating without any filler was called WPUA.

The tinplate was sanded evenly with 200 μm sandpaper until the surface has a certain roughness, and then the surface was cleaned with alcohol. A 150 μm wire rod applicator was used to prepare a coating film on the treated tinplate. The films were formed at room temperature and dried in a blast oven at 40 °C for 7 days to obtain the final sample.

**Figure 2.** Reaction scheme to synthesize waterborne polyurea (WPUA), GO/WPUA, and SG/WPUA coatings.

*2.5. Characterization*

The Fourier transform infrared spectrum (FT-IR, Nicolet 6700, Thermo, Waltham, MA, USA) of the sample was recorded in the wavenumber range of 4000–500 cm$^{-1}$. X-ray diffraction (XRD, Bruker Co., Karlsruhe, Germany) chart was recorded with Cu K$_\alpha$ radiation source ($\lambda$ = 0.15406 nm) to analyze the structure of samples. The microstructure of the sample was observed by a scanning electron microscope (SEM, Hitachi, Tokyo, Japan), and the dried film was quenched with liquid nitrogen. The $^1$H nuclear magnetic resonance ($^1$H NMR, Bruker 600 MHz NMR spectrometer, Karlsruhe, Germany) spectrum was based on d$_6$-DMSO as the solvent and the residual proton impurities in DMSO at $\delta$ = 2.50 ppm as a reference.

The particle size distribution of WPUA emulsions was conducted by dynamic light scattering (DLS, Zetasizer Nano, Malvern Instruments Limited, Malvin, UK). The water contact angle was measured by a contact angle goniometer (JC2000D2A, Shanghai Zhongchen Digital Technology Equipment Co., Ltd., Shanghai, China). The film was placed in deionized water for 24 h, then taken out and dried with filter paper. The water absorption rate is calculated according to the following Equation (1) [22]:

$$W = \frac{m_1 - m_0}{m_0} \times 100\% \tag{1}$$

where the mass of the film before and after immersion is $m_0$ and $m_1$, respectively.

An electrochemical workstation (CHI660E, Shanghai Chenhua Equipment Co., Ltd., Shanghai, China) was used to characterize the protective properties of the coatings. A three-electrode system was used, with a saturated calomel electrode (SCE) as the reference electrode, a platinum plate as the counter electrode, and a test sample as the working electrode. The potentiodynamic polarization curves were obtained by scanning from cathodic to the anodic direction ($E_{OCP} \pm 300$ mV). In addition, the electrochemical impedance spectroscopy (EIS) measurement frequency range was $10^{-2}$–$10^5$ Hz. The working electrode with an area of 1 cm$^2$ was exposed to a 3.5 wt.% NaCl solution and immersed for different times (6, 24, and 72 h). All tests were performed under environmental conditions and repeated three times to ensure that all data are repeatable.

## 3. Results and Discussion

### 3.1. Characterization of SG and WPUA

The FT-IR spectra of graphene oxide (GO) and sulfonated graphene (SG) were obtained by the KBr tableting method shown in Figure 3a. The infrared spectrum of GO confirmed the existence of C–O ($\nu_{C-O}$ at 1043 cm$^{-1}$), C–O–C ($\nu_{C-O-C}$ at 1101 cm$^{-1}$), and C–OH ($\nu_{C-OH}$ at 1369 cm$^{-1}$); the absorption peak at 1723 cm$^{-1}$ comes from the C=O tensile vibration on the carbonyl and carboxyl groups. The strong peak in the range of 3434–3330 cm$^{-1}$ is the vibrational absorption peak of –OH, which is attributed to the oxidation of GO. The absorption band of about 1618 cm$^{-1}$ caused by the skeleton vibration of the unoxidized graphite domain. For sulfonated graphene (SG), the peaks at 1219, 1154, and 1069 cm$^{-1}$ (two $\nu_{S-O}$ and one $\nu_{S-phenyl}$) indicate the presence of sulfonic acid groups. The peaks at 1036 cm$^{-1}$ ($\nu_{C-H}$ in-plane bending) and 729 cm$^{-1}$ (out-of-plane hydrogen wagging) are characteristic vibrations of a p-disubstituted phenyl group [27]. The weakened –OH absorption peak (3434–3330 cm$^{-1}$) of SG is due to the reduction in some oxygen-containing groups. In addition, the absorption peaks at 1049, 1101, and 1369 cm$^{-1}$ are severely weakened, demonstrating that the epoxide and hydroxyl groups attached to the graphene nanosheets have been reduced. Even though the SG undergoes a reduction procedure, the peak at 1709 cm$^{-1}$ still shows the presence of residual C=O on the nanosheet. The carbonyl group is believed to come from the acid moiety located on the edge of the graphene sheet [28].

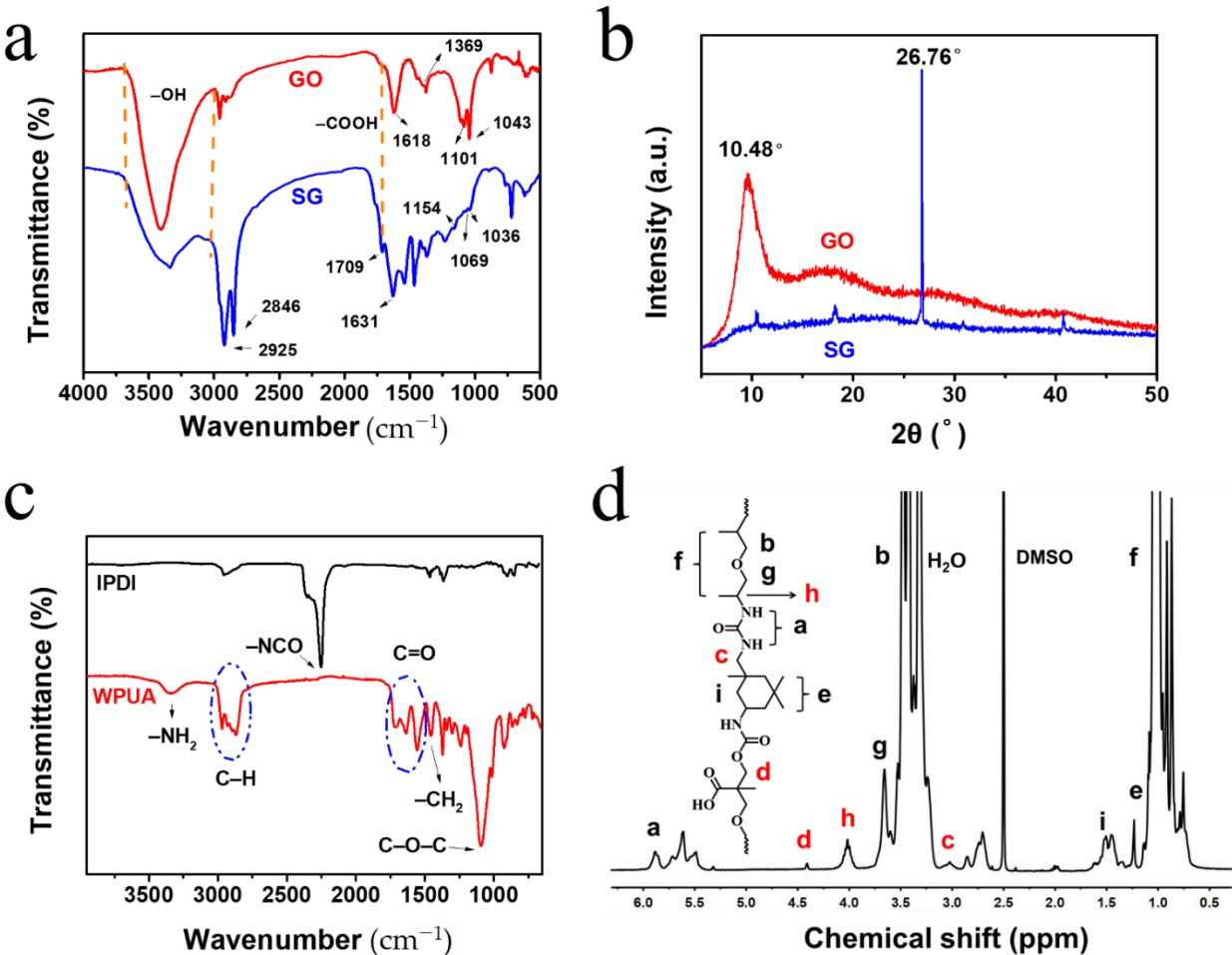

**Figure 3.** FT-IR spectra (**a**) and X-ray diffraction spectra (**b**) of GO and SG, FT-IR spectra (**c**), and [1]H NMR spectrum (**d**) of WPUA.

The XRD pattern is shown in Figure 3b. GO and SG show diffraction peaks at 10.48° and 26.76°, respectively. The interlayer spacing of GO and SG nanosheets is calculated to be about 0.8435 and 0.3424 nm. This shows that during the preparation of GO from graphite, the increase of oxygen-containing groups between the layers of the sheets makes the graphene layer spacing increase and the stacking disorder. In addition, some oxygen-containing functional groups on the surface of the SG nanosheets are reduced, which narrows the distance between the graphene sheets. The interlayer distance of SG (0.3424 nm) is close to the theoretical value of graphite interlayer distance (about 0.3440 nm), which indicates that most of the sulfonic acid groups are grafted on the edge of the graphene sheet instead of on the surface [29].

As shown in Figure 3c, the FTIR comparative spectra of WPUA and IPDI were measured. The special absorption peak near 2253 $cm^{-1}$ belongs to the –NCO group of IPDI, and the –NCO group absorption band of the WPUA film disappears. The absorption bands around 1629 $cm^{-1}$ (C=O stretching vibrations of urea and DMPA carboxylate groups) demonstrated the existence of urea linkage. Simultaneously, a strong peak appears at 1722 $cm^{-1}$ (C=O stretching) and 1556 $cm^{-1}$ (N–H stretching vibrations) attributed to the urethane groups generated by the reaction of isocyanate and the hydroxyl group of DMPA. In addition, the N–H bond vibration peaks of the primary amines of the polyurea macromolecules are displayed at about 3346 $cm^{-1}$. The absorption peak at 1344 $cm^{-1}$ was attributed to C–N, and the strong absorption peak of the ether bond on polyetheramine is shown at 1092 $cm^{-1}$.

The $^1$H NMR spectrum of the prepared polyurea is shown in Figure 3d, and the assignment of each peak is as follows: The 1.58 and 1.23 ppm peaks are derived from the –CH$_3$ and –CH$_2$ protons on the cyclohexyl group of the IPDI unit, respectively. The proton peaks of 3.08 and 4.02 ppm are, respectively, attributed to –CH$_2$ (IPDI unit) and –CH (polyetherimide segment), which are located near the urea group. The proton peaks at 1.04, 3.32, and 3.66 ppm belong to –CH$_3$, –CH$_2$, and –CH$_2$ (near the urea group) of the polyether amine chain, respectively. A weak peak of methylene protons (DMPA moiety) close to urethane was observed at 4.40 ppm, and the protons on the urea group appeared at a chemical shift value of about 5.89 ppm, which appeared as a very weak broad peak. Combined with FTIR characterization, these results further confirmed the successful preparation of water-borne polyurea.

### 3.2. Dispersion Stability of GO and SG in WPUA

The average particle size of WPUA nanoparticles in different pH values (9–4, 25 °C) was evaluated. It was found that the average particle size increases as the pH values decreases, the emulsion flocculates until the pH = 3–4. We also simply evaluated the stability of WPUA emulsion at different temperatures (−10–40 °C). In addition, found that the viscosity of WPUA emulsion increased significantly at low temperatures (about −6−−10 °C). Stormer viscosity value: 76 KU (40 °C)–87 KU (−10 °C). This may be caused by the glass transition of the polyurea molecule. From Figure 4a and Table 1, the average particle size ($d$ = 171–179 nm, pH = 7–8, 25 °C) of the WPUA emulsion slightly changed after being placed for 7, 14, and 28 days, respectively. The particle size is distributed from 75 to 475 nm, and the peak appears at 191.7 to 195.5 nm. The smaller the PDI value (0.065–0.109), the narrower the particle size distribution and the more uniform the particle size. These prove that the prepared WPUA has excellent dispersibility and stability in water (pH = 7–8, 25 °C).

**Table 1.** WPUA particle size distribution data.

| Time (days) | Z-Ave (nm) | PDI | Peak (nm) |
|:---:|:---:|:---:|:---:|
| 7 | 177.7 | 0.065 | 191.7 |
| 14 | 178.3 | 0.089 | 194.3 |
| 28 | 177.1 | 0.109 | 195.5 |

The ζ potentials of GO and SG aqueous dispersions are about −32 and −46 mV at pH = 2.8–3.0, respectively. Both GO and SG can be stably dispersed in water, but the ζ potential of SG is more negative than GO, which is attributed to the stronger electrostatic repulsion between adjacent SG nanosheets.

The appearance of different coatings after standing for 28 days is illustrated in Figure 4b. The GO precipitation occurs in the GO/WPUA, and its proportion matches the amount of GO incorporated. Correspondingly, SG/WPUA did not find any sedimentation and the sample showed a uniform brown liquid state. The SEM images in Figure 4c show the nanoscopic dispersion state of GO and SG nanosheets in the WPUA matrix. The bright white lines and grey area represent the GO/SG nanosheets and the waterborne polyurea matrix, respectively. The stacking of the GO layer in the WPUA matrix is tortuous and disorderly, while the SG sheets have a parallel distribution in the WPUA matrix. The dissociation degree and electrostatic repulsion of sulfonic acid groups on SG nanosheets are higher, which makes it more compatible and dispersible in WPUA emulsion than GO. The ζ potential values of GO and SG aqueous dispersions also support this result.

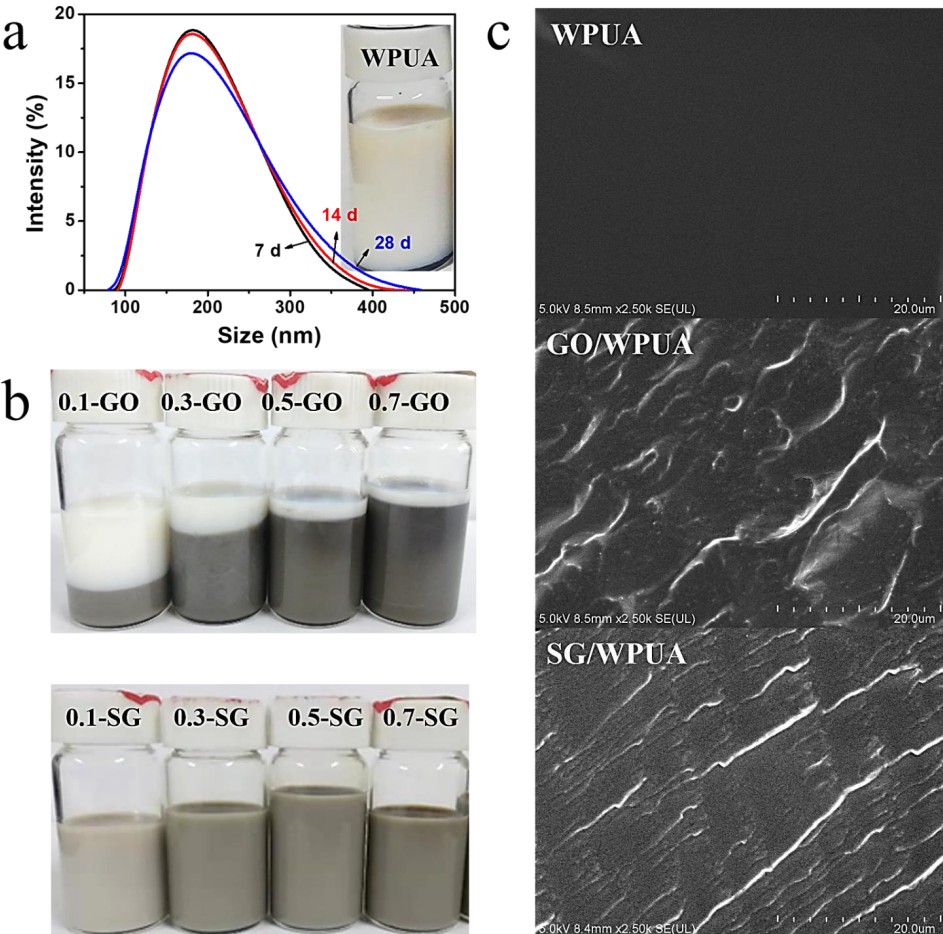

**Figure 4.** The particle size distribution of WPUA emulsion after standing for 7, 14, and 28 days (**a**); GO/WPUA and SG/WPUA coatings after 28 days of standing (**b**); SEM micrograph (**c**) of WPUA, GO/WPUA, and SG/WPUA with 0.3 wt.% content of filler.

On the other hand, similar phenomena were observed in the mixture of GO and other polymers with active hydrogen groups [27,30,31]. For example, hydroxyl-rich polyvinyl alcohol (PVA) chain can interact with two or more GO sheets by hydrogen bonds, forming a sufficient number of cross-linking sites, GO composite hydrogel is generated due to the formation of a GO network [32]. In the externally emulsified epoxy emulsion (WPE), hydrogen bonds are formed between GO and the surfactant, leading to the gelation and demulsification of the epoxy emulsions [29]. The mechanism is explained as follows: plenty of oxygen-containing groups on the surface of the GO sheets, forming hydrogen bonds with the urea groups on the polyurea chain. When the hydrophilic structure on the polyurea chain is insufficient to support the stable suspension of the two in water, this will cause the sedimentation of GO and part of the polyurea nanoparticles. There are fewer hydrogen bonds between SG and polyurea, which is beneficial to the storage stability of WPUA emulsion. In Figure 5, the dissociation mechanism of sulfonic acid groups on SG nanosheets in WPUA emulsion can be seen. The WPUA form hydrogen bonds with sulfonic acid groups (O=S=O) of SG nanosheets, rather than the ionic bond. In addition, the sulfonic acid groups (S–O$^-$) form ionic bond with TEA (neutralizer in WPUA emulsion).

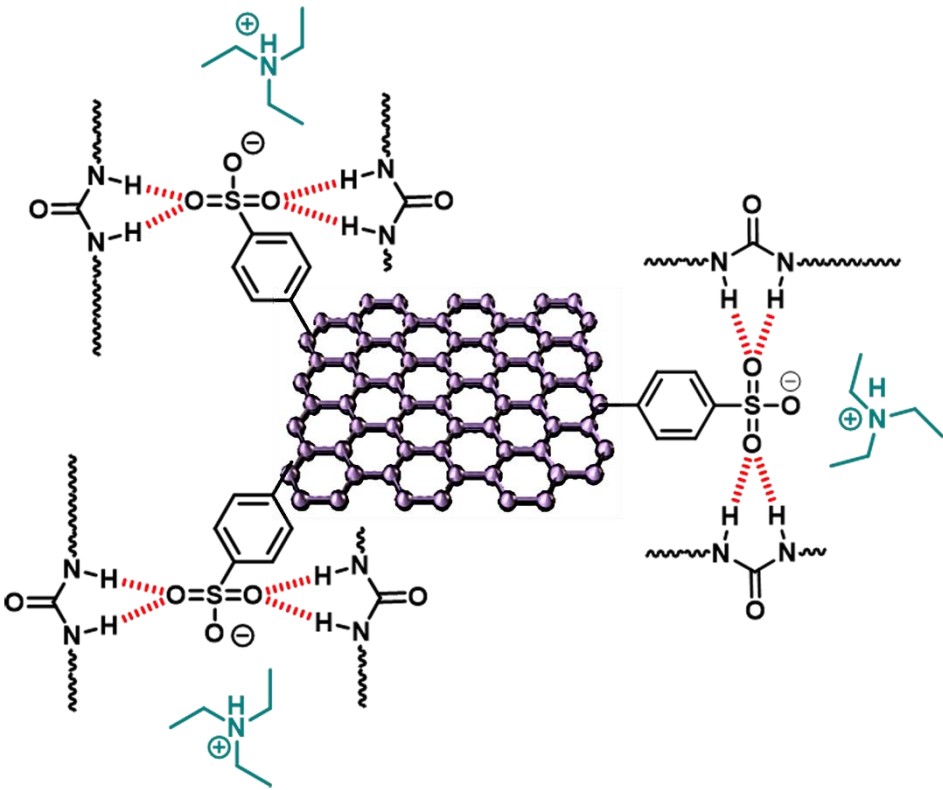

**Figure 5.** The dissociation mechanism of sulfonic acid groups on SG nanosheets in WPUA emulsion.

In the subsequent work, we will focus on the related testing and characterization of SG and its composite coating instead of GO, due to the poor dispersion stability of GO in WPUA.

### 3.3. Water Resistance of Composite Films

Figure 6a,b exhibit the water contact angle and water absorption for composite films with different contents of SG sheets. The water contact angle increases first and then decreases as the content of SG increases, while the water absorption decreased first and then increased. These phenomena suggest that the barrier properties of WPUA nanocomposite film were enhanced with the incorporation of SG nanosheets. The addition of a small amount (≤0.3 wt.%) of SG can enhance the hydrophobicity of the composite film. However, when the SG content increased to 0.7 wt.%, the performance decreased, which may be attributed to the weakened phase compatibility and increased phase aggregation.

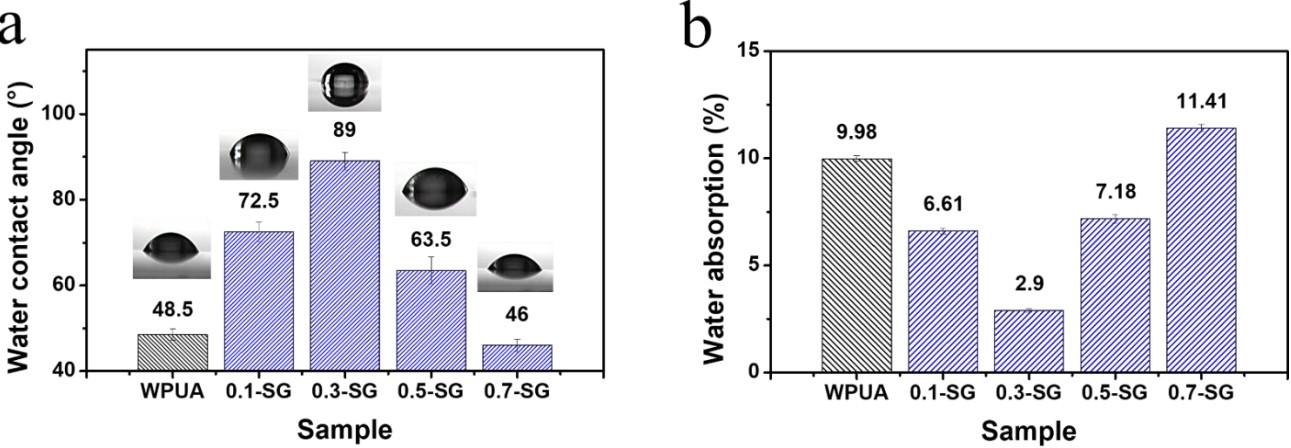

**Figure 6.** Water contact angle (**a**) and water absorption capacity (**b**) of different composite films.

The micrograph of 0.3-SG with the best water resistance and WPUA was selected for observation. From Figure 7a,c, the WPUA and 0.3-SG all showed black micro-cracks, which were generated by electrostatic repulsion between the anionic polyurea nanoparticles during the film-forming. The WPUA and 0.3-SG films were immersed in deionized water, then dried in air and quenched with liquid nitrogen, and the film's longitudinal section was observed by SEM. As shown in Figure 7b,d, these black circles represent the holes left by water infiltration, and the black lines mean micro-cracks. WPUA showed more holes with a size of about 150~400 nm. The 0.3-SG was found to have more enlarged micro-cracks with an average size of 100 nm, instead of holes. The phenomena can be explained as follows: $H_2O$ molecules penetrate the interior of the coating through the micro-cracks, and with the extension of the immersion time, the micro-cracks will further expand into larger holes. The existence of SG nanosheets will block the penetration of $H_2O$ and effectively prolong the infiltration path of $H_2O$. These prove that the uniform dispersion of SG nanosheets in WPUA can block the inherent shrinkage pores and defects of the coating and improve the water-resistance of the coating.

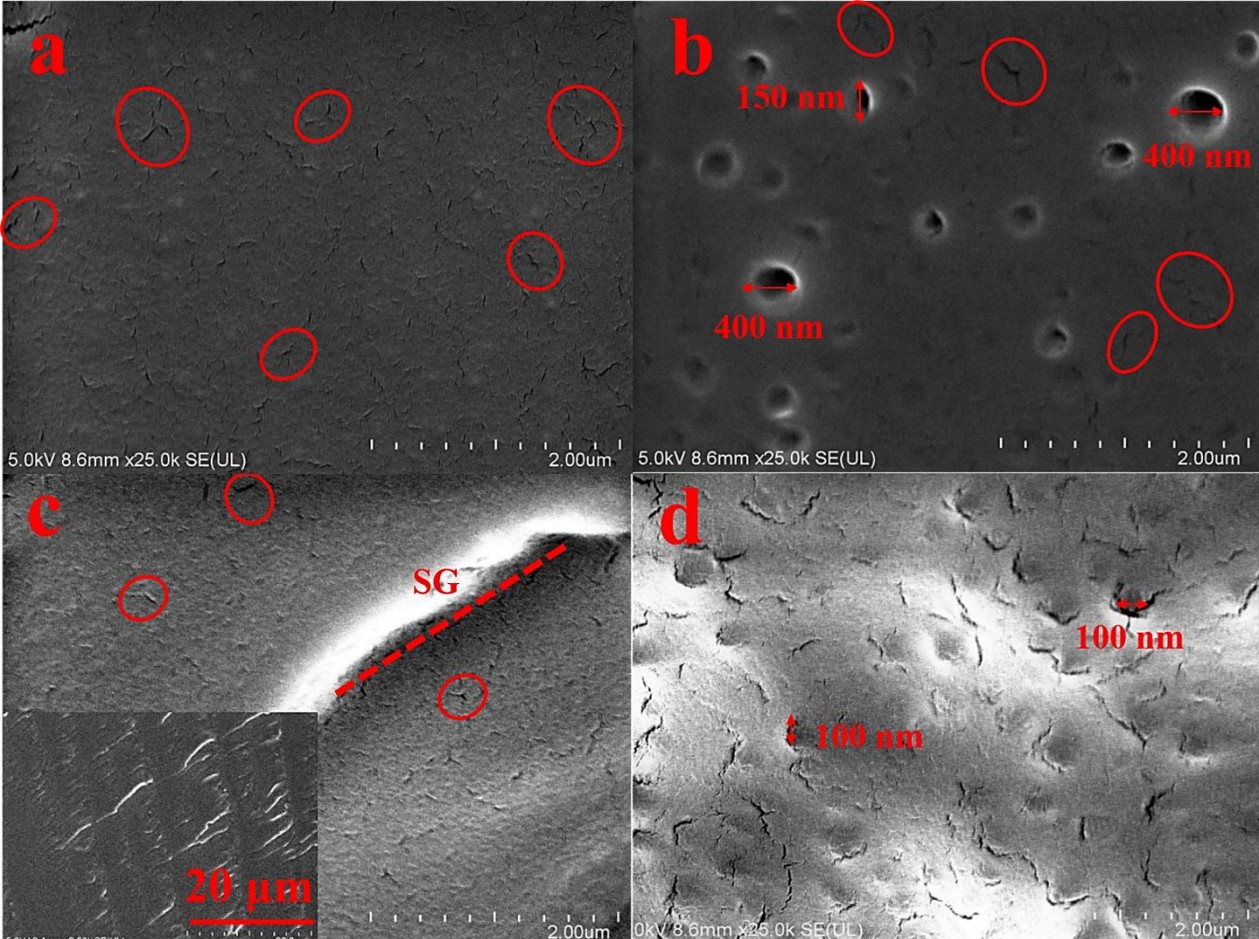

**Figure 7.** Micrographs of WPUA (**a**) and 0.3-SG (**c**) without immersion, and WPUA (**b**) and 0.3-SG (**d**) after immersion treatment with deionized water.

### 3.4. Corrosion Measurements

Potentiodynamic polarization curves and electrochemical impedance spectroscopy (EIS) tests were used to study the effect of SG content on the performance of polyurea composite coatings. As shown in Figure 8a and Table 2, the polarization curve shows that both the anode and cathode regions have a reasonably linear Tafel region [33]. In general, a lower $i_{corr}$ and a higher $E_{corr}$ indicate better anti-corrosion properties. The sequence

of $i_{corr}$ is 0.3-SG (0.3192 μA/cm$^2$) < 0.1-SG (0.8340 μA/cm$^2$) < 0.5-SG (0.9018 μA/cm$^2$) < WPUA (0.9772 μA/cm$^2$) < 0.7-SG (1.5834 μA/cm$^2$). Correspondingly, the $E_{corr}$ of the WPUA and SG/WPUA is 0.7-SG (−0.721 V) < WPUA (−0.678 V) < 0.5-SG (−0.609 V) < 0.1-SG (−0.533 V) < 0.3-SG (−0.508 V). The 0.3-SG has a more effective anti-corrosion effect, and the 0.7-SG is worse than WPUA. This indicates that the addition of 0.7 wt.% has exceeded the penetration threshold of the SG filler [34], which has a negative impact on coating protection. In addition, the corrosion rate ($v_{corr}$) of 0.3-SG (0.2223 × 10$^{-6}$ g/h) is the lowest among all SG/WPUA samples, especially the blank sample WPUA reaches 0.8632 × 10$^{-6}$ g/h. The addition of 0.3 wt.% SG improves the protection efficiency (η) of the composite coating to the optimal value of 67.34%, showing the best corrosion inhibition performance.

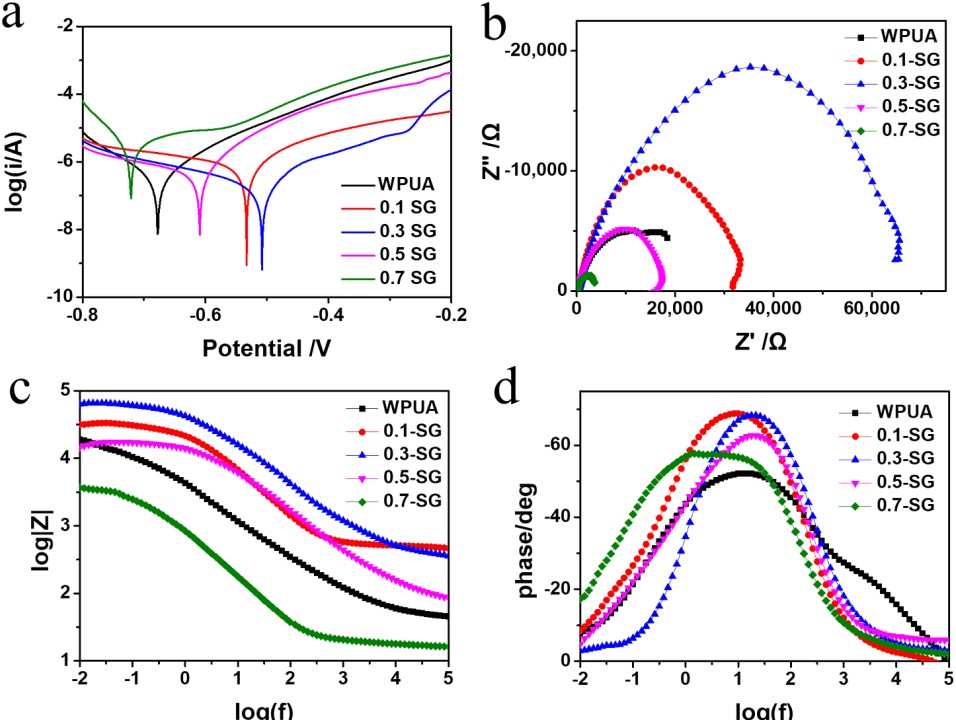

**Figure 8.** Tafel plots (**a**), Nyquist plots (**b**), Bode plots (**c**) and (**d**) of WPUA, 0.1-SG, 0.3-SG, 0.5-SG and 0.7-SG immersed in 3.5% NaCl aqueous solution.

**Table 2.** Tafel electrochemical measurement data.

| Samples | $E_{corr}$ (V) | $i_{corr}$ (μA/cm$^2$) | $v_{corr}$ (g/h) | η (%) |
|---|---|---|---|---|
| WPUA | −0.678 | 0.9772 | 0.8632 × 10$^{-6}$ | — |
| 0.1-SG | −0.533 | 0.8340 | 0.3091 × 10$^{-6}$ | 14.65 |
| 0.3-SG | −0.508 | 0.3192 | 0.2223 × 10$^{-6}$ | 67.34 |
| 0.5-SG | −0.609 | 0.9018 | 0.5147 × 10$^{-6}$ | 7.72 |
| 0.7-SG | −0.721 | 1.5834 | 2.534 × 10$^{-6}$ | −62.03 |

$E_{corr}$—corrosion potential; $i_{corr}$—corrosion current density; $v_{corr}$—corrosion rate; η—protection efficiency [35].

The parameters reflecting the corrosion protection performance of the coating were calculated according to Equation (2):

$$\eta\% = \left[1 - \frac{i_{corr,WPUA}}{i_{corr,SG/WPUA}}\right] \times 100\% \qquad (2)$$

In Equation (2), $i_{\mathrm{corr,WPUA}}$—corrosion current density of WPUA sample, $i_{\mathrm{corr,SG/WPUA}}$—corrosion current density of SG/WPUA samples (includes 0.1-SG, 0.3-SG, 0.5-SG, 0.7-SG samples).

As can be seen from Figure 8b, the diameters of these semicircles vary with different additives in Nyquist plots. It can also be found that the addition of 0.1 wt.% and 0.3 wt.% SG nanosheets increases the diameter of the semicircle significantly, indicating that SG nanosheets can improve corrosion protective performance of pure water-borne polyurea resin coating.

Bode curves of the sample are presented in Figure 8c. The impedance modulus values of WPUA coating are in the lowest position compared with those of other coatings (except 0.7-SG) with different content of additives. From Figure 8d, generally, the phase angle values at intermediate frequency ranges ($10^4$–$10^5$ Hz) are the characteristic response of the coating defect. The single-phase peak in the mid-frequency region occupies the dominant position, and corrosion can be evaluated by comparing phase angle values of the peak. When the different content of SG nanosheets was added to water-borne polyurea resin, the phase angles obviously became larger, among which the phase angle of 0.3-SG and 0.1-SG coating is almost to $-70°$ at intermediate ($10^3$–$10^0$ Hz) frequencies, compared with WPUA coating ($-50°$).

To further investigate the effects of different immersion times (6, 24, and 72 h, respectively) on the corrosion protective performance of composite coatings with different content of SG nanofiller, the Bode impedance plots were further analyzed. The impedance modulus $|Z|$ at the lowest frequency ($f = 0.01$ Hz) in the Bode impedance curve represents the ability of the coating to impede the flow of current between the cathode and anode. $|Z|_{f = 0.01\ \mathrm{Hz}}$ can be regarded as a semi-quantitative index of corrosion resistance to evaluate the protective performance of the coating. The $\log |Z|_{f = 0.01\ \mathrm{Hz}}$-timeline is shown in Figure 9. The $|Z|_{f = 0.01\ \mathrm{Hz}}$ value of WPUA coating was about $1.9 \times 10^4\ \Omega \cdot \mathrm{cm}^2$ at 6 h immersion, with the prolongation of the exposure time, the $|Z|$ value of the WPUA coating dropped to $1.4 \times 10^3\ \Omega \cdot \mathrm{cm}^2$ at 136 h. It demonstrates that the WPUA coating has poor barrier properties to corrosive electrolytes. For 0.3-SG samples, the value of $|Z|_{f = 0.01\ \mathrm{Hz}}$ has a range of $6.4 \times 10^4$ to $2.2 \times 10^4\ \Omega \cdot \mathrm{cm}^2$, which was higher than that of the WPUA samples during the whole immersion process. Its remarkable protective performance is attributed to 0.3 wt.% of SG nanosheets, which are uniformly dispersed in polyurea resin and can resist the diffusion of corrosive electrolyte.

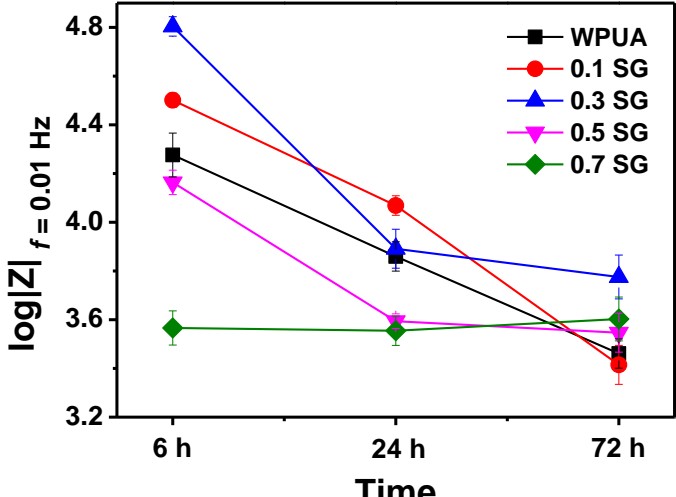

**Figure 9.** The variation on the impedance modulus ($|Z|_{f = 0.01\ \mathrm{Hz}}$) for the different coating samples as a function of immersion time.

From Figure 10a,b, Nyquist plots of WPUA and 0.3-SG were fitted with an electrical equivalent circuit and utilized to fit corrosion parameters to study the anti-corrosion mechanism of SG nanosheets in polyurea systems. The models in Figure 10c,d were used to match the EIS data with an immersion time of 6, 24, and 72 h, respectively, and the electrochemical parameter data are shown in Table 3, where $R_s$, $R_{ct}$, $R_c$, $C_c$, $CPE_{dl}$, and $CPE_c$ are the solution resistance, charge transfer resistance, coating resistance, coating capacitance, constant phase element of double layer, and constant phase element of the coating, respectively. The capacitance value of $CPE_{dl}$ was calculated by Equation (3) [36]:

$$C = Y_0(\omega_{max})^{n-1} \tag{3}$$

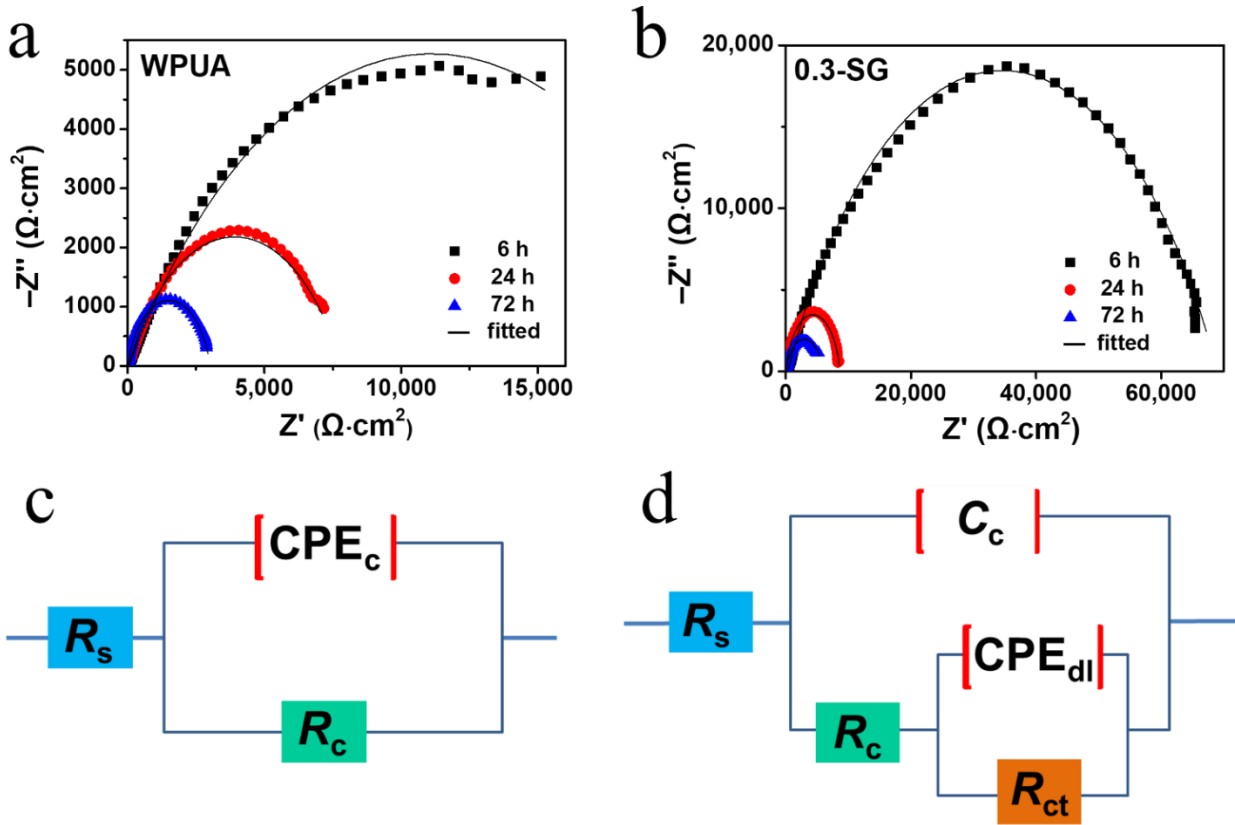

**Figure 10.** Nyquist plots of WPUA (**a**) and 0.3-SG (**b**); electrical equivalent circuit diagrams fitting Nyquist plots for immerse to 6 h (**c**), 24 h, and 72 h (**d**).

**Table 3.** The electrochemical parameter data of WPUA and 0.3-SG samples immersed for 6, 24, and 72 h.

| Time | Sample | CPE$_c$ | | | $R_c$ ($\Omega \cdot cm^2$) | CPE$_{dl}$ | | | $R_{ct}$ ($\Omega$ cm$^2$) |
|------|--------|---------|---|---|---|---------|---|---|---|
| | | $Y_0$ ($\Omega^{-1} \cdot cm^{-2} \cdot s^n$) | $n$ | $C_c$ (F·cm$^{-2}$) | | $Y_0$ ($\Omega^{-1} \cdot cm^{-2} \cdot s^n$) | $n$ | $C_{dl}$ (F cm$^{-2}$) | |
| 6 h | WPUA | $7.55 \times 10^{-5}$ | 0.57 | $1.18 \times 10^{-5}$ | $2.15 \times 10^2$ | — | — | — | — |
| | 0.3-SG | $4.16 \times 10^{-7}$ | 0.63 | $1.95 \times 10^{-7}$ | $6.96 \times 10^4$ | — | — | — | — |
| 24 h | WPUA | — | — | $1.20 \times 10^{-5}$ | 2.70 | $9.80 \times 10^{-5}$ | 0.66 | $9.04 \times 10^{-5}$ | $7.74 \times 10^3$ |
| | 0.3-SG | — | — | $1.39 \times 10^{-6}$ | 15.15 | $1.76 \times 10^{-5}$ | 0.86 | $1.25 \times 10^{-5}$ | $8.61 \times 10^4$ |
| 72 h | WPUA | — | — | $1.67 \times 10^{-5}$ | 1.82 | $2.32 \times 10^{-4}$ | 0.80 | $2.19 \times 10^{-4}$ | $3.01 \times 10^3$ |
| | 0.3-SG | — | — | $8.39 \times 10^{-6}$ | 3.73 | $1.19 \times 10^{-4}$ | 0.75 | $1.07 \times 10^{-4}$ | $9.54 \times 10^3$ |

Among them, $Y_0$ is the constant of CPE, and $\omega_{max}$ (rad·s$^{-1}$) corresponds to the angular frequency where the imaginary part of the impedance is maximum. The parameter $n$ ($0 < n < 1$) is an exponential term. When $n = 0$, CPE$_{dl}$ is pure resistance; when $n = 1$, CPE$_{dl}$ is pure capacitance [37].

The speed of electrolyte penetration is an important indicator to evaluate the barrier performance of the coating, which will change the coating capacitance ($C_c$) and coating pore resistance ($R_c$). As the immersion time increases, the $C_c$ value increases with electrolyte penetration. The $C_c$ value of WPUA is about 2-to-1 orders of magnitude higher than 0.3-SG, which shows that the electrolyte penetrates WPUA faster. Moreover, the $R_c$ value of 0.3-SG is always greater than that of WPUA, which was attributed to the combination of incorporation of flake-like SG reduced coating defects. The corrosive medium occupied the microcracks of the WPUA matrix between the SG nanosheets and penetrated the coating in a tortuous manner. In addition, the electrochemical corrosion process at the metal interface consisted of an $R_{ct}$ and a CPE$_{dl}$. It can be seen that the $R_c$ values decreased slowly from $7.74 \times 10^3$ to $3.01 \times 10^3$ $\Omega$·cm$^2$ as the immersion time was prolonged to 72 h, while the $R_{ct}$ values of 0.3-SG coating decreased from $8.61 \times 10^4$ to $9.54 \times 10^3$ $\Omega$·cm$^2$ with the immersion time was prolonged from 24 to 72 h, reflecting that the corrosion resistance of 0.3-SG was better than pure WPUA in total.

The polarization curve and EIS test results indicate that excess graphene sheets are more prone to stacking and agglomeration, which reduces the specific surface area of the nanomaterial. With the large size of the re-stacking or agglomerating particles, they may form defect points in the coating. The low-content ($\leq$0.03 wt.%) SG can be more uniformly dispersed and preferably dispersed in the matrix at the molecular level [27], which increases the diffusion length of the etchant through the micropores to the defects. As soon as the graphene sheet forms a physical barrier network, waterproof performance and corrosion resistance can be improved.

## 4. Conclusions

In this work, SG was prepared by covalently grafting sulfonic acid groups to the edges of graphene nanosheets. Self-emulsified WPUA was synthesized, and the polyurea chain consists of hard segment (IPDI), soft segment (D2000, D230), and anionic internal dispersant (DMPA for electrostatic stabilization). We then incorporated the SG nanosheets into the WPUA to prepare composite coating (SG/WPUA).

The dispersion stability, water resistance, and corrosion resistance of the composite coating were systematically characteristic analyzed. The results show: (1) SG sheets have a parallel distribution in the WPUA matrix from the SEM test. SG has a greater compatibility and dispersion in WPUA emulsion and is stable for a long time (28 days) without delamination. (2) The 0.3 wt.% SG nano-filler significantly improves the waterproof performance of the composite coating; water contact angle (WCA) result: SG/WPUA (89°) > WPUA (48.5°), water absorption result: SG/WPUA (2.90%) < WPUA (9.98%). After water immersion treatment of the film, SEM observed that the SG/WPUA film only generated enlarged microcracks (100 nm) instead of holes (150–400 nm, WPUA film). (3) The polarization curve and EIS test show that the composite coating with 0.3 wt.% SG has the highest corrosion-resistance efficiency. Combining the advantages of sulfonated graphene and polyurea, it can provide a theoretical basis for the application of sulfonated graphene/waterborne polyurea (SG/WPUA) composite coatings in harsh environments.

**Author Contributions:** Conceptualization, J.Z. and J.W. (Jihu Wang); methodology, J.Z.; visualization, J.Z.; validation, S.L.; formal analysis, C.W. and J.W. (Jihu Wang); data curation, Y.C. and Y.W.; writing—original draft preparation, J.Z.; writing—review and editing, J.Z. and J.W. (Jing Wang); supervision, J.W. (Jihu Wang) and S.W.; project administration, S.W.; funding acquisition, X.Y. and Y.M. All authors have read and agreed to the published version of the manuscript.

**Funding:** The research was supported by the Special Projects of the Ministry of Science and Technology Key R&D Program (Grant No. 2018YFC1801503).

**Institutional Review Board Statement:** Not applicable.

**Informed Consent Statement:** Not applicable.

**Data Availability Statement:** The data presented in this study are available in article.

**Conflicts of Interest:** The authors declare no conflict of interest.

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
