# Peer review of "Waterborne Polyurea Coatings Filled with Sulfonated Graphene Improved Anti-Corrosion Performance"

_coatings, doi:10.3390/coatings11020251_

Round 1
Reviewer 1 Report
This article reports the anti-corrosion performance of polyurea-based coatings containing graphene derivatives.
The represented data are well-organized, and can contribute to the practical aspects of corrosion.
I can recommend for publication, after some minor considerations as following.
1) At line 116-117, if Figure 2 is correct, the sentence should be modified:
“The product was dispersed in deionized water at high speed (1500 rpm) for 30 minutes using a disperser.”
2) At line 123-124, “sulfonated graphene/water-borne polyurea (GO/WPUA)” should be modified to “graphene oxide/water-borne polyurea (GO/WPUA)”.
3) At line 150, a word, “battery” is not needed. Authors used “A three-electrode system”.
4) At line 152-153, the description is confusing, because frequency range is meaningful only for electrochemical impedance spectroscopy. Please modify here.
5) At equation 2, please confirm the definition. Readers cannot reproduce the protection efficiency in Table 2.
Author Response
Dear editor and reviewers,
Thanks a lot for the valuable suggestions and instructions on improving the manuscript. We have studied the comments carefully and have made the corresponding corrections. The email address of the authors was revised to the institutional email address, referring to suggestions from the editorial department. We use the "Track Changes" function in Microsoft Word, and revised parts are marked in yellow in the manuscript. The responses to the reviewer comments are as following:
Response to Reviewer 1 Comments
1. At line 116-117, if Figure 2 is correct, the sentence should be modified: “The product was dispersed in deionized water at high speed (1500 rpm) for 30 minutes using a disperser.”
Thanks! it has been modified at line 116-118.
2. At line 123-124, “sulfonated graphene/water-borne polyurea (GO/WPUA)” should be modified to “graphene oxide/water-borne polyurea (GO/WPUA)”.
Thanks a lot for pointing it out. it has been modified at line 124.
3. At line 150, a word, “battery” is not needed. Authors used “A three-electrode system”.
At line 149, the word “battery” has been deleted.
4. At line 152-153, the description is confusing, because the frequency range is meaningful only for electrochemical impedance spectroscopy. Please modify here.
At line 151-154, it has been revised.
5. At equation 2, please confirm the definition. Readers cannot reproduce the protection efficiency in Table 2.
We have revised it at lines 321-324, and pointed out a reference (doi.org/10.1016/j.porgcoat.2019.04.013). η-protection efficiency, as the parameters reflecting corrosion protection performance of the coating, it’s calculated by values (corrosion current densities) of the different samples.
Reviewer 2 Report
The paper entitled “Waterborne polyurea coatings filled with sulfonated graphene improved anti-corrosion performance” can be published after revision. The following issues must be addressed:
- The Introduction need more examples of films with which chemical and mechanical stability (e.g. DOI: 10.1016/j.matlet.2011.03.111, DOI: 10.1016/j.tsf.2010.12.199);
- The authors must outline in the Introduction part what is new and innovative in this work;
- Explain in more details why you consider that the oxygen-containing functional groups on the surface of the SG nanosheets are reduced in a certain amount.
- Have you evaluated the poly-urea dispersability and stability in other environments?
- Explain in more details the dissociation behavior of sulfonic acid groups on SG nanosheets.
- It will be interesting to include (at least in the future work) the composite films stability in saline conditions.
- The Conclusion part should include some informations about the composition and morphology of the samples.
Author Response
Dear editor and reviewers,
Thanks a lot for the valuable suggestions and instructions on improving the manuscript. We have studied the comments carefully and have made the corresponding corrections. The email address of the authors was revised to the institutional email address, referring to suggestions from the editorial department. We use the "Track Changes" function in Microsoft Word, and revised parts are marked in yellow in the manuscript. The responses to the reviewer comments are as following:
Response to Reviewer 2 Comments
1. The Introduction need more examples of films with which chemical and mechanical stability (e.g. DOI: 10.1016/j.matlet.2011.03.111, DOI: 10.1016/j.tsf.2010.12.199);
thanks for your suggestions, references have been added to the article at line 59.
2. The authors must outline in the Introduction part what is new and innovative in this work;
In the introduction part, we have made a detailed revision. The main purpose of the article is to propose a strategy: SG is applied to water-based polyurea to improve material properties. For this work, the dispersion and compatibility of the SG-doped WPUA matrix were characteristics analyzed, and the performance of the composite coating (including water resistance and corrosion resistance) was systematically evaluated.
3. Explain in more details why you consider that the oxygen-containing functional groups on the surface of the SG nanosheets are reduced in a certain amount.
We have revised it at lines 63-66 in the introduction. Herein, we make a brief explanation:
Graphene has impermeable properties, due to its conjugated surface structure, small molecules cannot penetrate. The oxidative modification destroys the honeycomb structure on the surface of graphene, and the graft modification on GO further destroys it, causing defects on the surface of graphene. Therefore, it is necessary to reduce the oxygen-containing groups to eliminate partial defects on the SG surface.
In the “3.1.Characterization of SG and WPUA” section, the prepared SG and GO are analyzed by FTIR and XRD tests, we found that the IR absorption peaks of -OH, -COOH, and -O- of SG were significantly weakened. Moreover, the SG sheet spacing is reduced to 0.3424 nm, which is close to the theoretical sheet spacing of graphite, indicating that the oxygen-containing groups on both sides of the SG sheet have been reduced by a certain amount.
4. Have you evaluated the poly-urea dispersability and stability in other environments?
At lines 213-219. We have evaluated the particle size of polyurea nanoparticles in different pH values, and the stability of WPUA emulsion at different temperatures (-10°C ~ 40°C).
5. Explain in more details the dissociation behavior of sulfonic acid groups on SG nanosheets. It will be interesting to include (at least in the future work) the composite films stability in saline conditions.
thank you very much for your comment! We added Figure 5 at line 267. “The dissociation mechanism of sulfonic acid groups on SG in WPUA emulsion” is explained at line 259-263.
In our view, the salt ions will form ionic bond with the sulfonic acid group (S-O-), and it does not destroy the hydrogen bonds. So, salt ions can not affect the stability of composite films.
6. The Conclusion part should include some informations about the composition and morphology of the samples.
thank you for your kind suggestions! We have modified this part, referring to your suggestion.
Round 2
Reviewer 2 Report
The manuscript can be published in the prezent form
This manuscript is a resubmission of an earlier submission. The following is a list of the peer review reports and author responses from that submission.